# Mutually Supportive and Inclusive Societies Driven by Community Social Workers in Japan: A Thematic Analysis of Japanese Comics

**DOI:** 10.3390/geriatrics8060113

**Published:** 2023-11-18

**Authors:** Ryuichi Ohta, Yumi Naito, Chiaki Sano

**Affiliations:** 1Community Care, Unnan City Hospital, 96-1 Iida, Daito-cho, Unnan 699-1221, Japan; somayum38@gmail.com; 2Department of Community Medicine Management, Faculty of Medicine, Shimane University, 89-1 Enya-cho, Izumo 693-8501, Japan; sanochi@med.shimane-u.ac.jp

**Keywords:** social isolation, community social workers, inclusive communities, aging societies, vulnerable populations, social disparities

## Abstract

Social isolation is a growing concern worldwide, particularly within aging populations. This study elucidates the specific attitudes, strategies, and approaches of community social workers (CSWs) in Japan as they work toward alleviating social isolation and building inclusive communities. This qualitative study, conducted in Toyonaka City, Osaka Prefecture, Japan, used six Japanese comics as a unique data source, narrating real-life stories of social isolation and CSWs’ approaches. Thematic analysis was conducted to analyze the content of the comics, including systematic coding, theme generation, and refining, while ensuring rigor and reflexivity. The total number of pages in the comics was 505. The transcripts of the comics as Microsoft Word documents totaled 63 pages. Four themes characterizing CSWs’ strategies were revealed: (1) core values of professionalism and dedication; (2) personalized support oriented toward person-centered suffering; (3) community engagement, transitioning from exclusiveness to inclusiveness; and (4) connecting and reorganizing communities for inclusive societies. In Japan, CSWs employ multifaceted strategies to combat social isolation and foster inclusive communities. Their dedication, personalized support, community engagement, and capacity to reorganize their communities contribute to their pivotal role. This study provides a foundation for understanding CSWs’ work and paves the way for further investigation of their evolving role in creating inclusive societies.

## 1. Introduction

Societies worldwide are witnessing an increase in social isolation, a phenomenon deeply rooted in the challenges posed by rapidly aging populations and ever-widening social disparities [1]. As societies age, traditional support networks weaken, leading to individuals, particularly older adults, becoming isolated from their communities [2]. Concurrently, social disparities exacerbate this isolation, as individuals from different socioeconomic backgrounds may find it challenging to connect or relate to one another [3]. Social disparities can also lead to poverty-related issues for older adults and children [4].

Addressing this phenomenon requires innovative solutions that not only identify but also reach out to those who find themselves stranded in the chasms between existing social systems [5]. One solution posits establishing and nurturing new relationships within communities or rebuilding previous social interactions with both locals and immigrants [6]. By forging new social ties, it becomes feasible to bridge the gaps in the social systems and combat increasing rates of social isolation [7,8]. Creating relationships in communities can enhance the social capital of the people and communities.

Community social workers (CSWs) have been identified as potential change agents in this context [9]. With their unique positioning and specialized training, CSWs can detect and address gaps in the social system, working toward remedying the issue of social isolation [10]. However, the effectiveness of CSWs in addressing this pressing issue is not solely based on their presence, but is intricately tied to the quality of their work and interventions.

Thus, understanding the exact nature, methodologies, and approaches of CSWs is crucial. The quality and efficacy of their interventions can substantially influence their ability to connect individuals and rebuild fractured community ties [11,12]. To date, no research has been conducted on the quality and efficacy of the social work performed by CSWs or their efforts to bridge and create relationships in communities.

Gaining clarity on the concrete approaches used by CSWs could provide critical insights into the strategies able to alleviate social isolation [13]. This study examines the specific attitudes, strategies, and approaches of CSWs in Japan in their battle against social isolation. Through this exploration, we hope to offer a comprehensive understanding that can guide future interventions and policies.

## 2. Materials and Methods

This qualitative study was performed using content from Japanese nonfictional comics describing real stories of social isolation and CSW techniques to clarify CSWs’ concrete approaches to alleviating social isolation in Japan. CSWs’ work quality and methods depend on their concrete working situations, clients, and work cultural context. Relativist ontology and constructivist epistemology perspectives can be used to better understand how Japanese CSWs perform their job while respecting the culture and contexts they encounter [14], an area that has not yet been examined. Thus, we conducted both inductive and reflective thematic analyses from these perspectives.

### 2.1. Setting: Toyonaka City

All Japanese CSW-related nonfiction comics used in this study described issues specific to Toyonaka City, Osaka. Toyonaka is located in Osaka Prefecture in the Kansai region of Japan. It is part of the larger Osaka metropolitan area and situated between Osaka City and the historical city of Ikeda. Toyonaka has a large population, a substantial proportion of which comprises older adults, making healthcare, welfare, and elderly care issues of prime importance. In 2020, 25.7% of the population comprised over 65-year-old people.

Similar to the rest of Japan, Toyonaka benefits from the universal healthcare system. All residents are required to enroll in either the National Health Insurance or Employee Health Insurance programs. Toyonaka, consistent with its urban setting, has many modern hospitals and clinics catering to various specialties. The welfare system in Toyonaka, reflective of Japan’s national standards, supports the economically vulnerable, disabled, and elderly. Given Japan’s rapidly aging population, elderly care has become a key national issue, and Toyonaka is no exception. Like many other urban areas in Japan, Toyonaka faces challenges with the increasing demand for elderly care, owing to its aging population. However, in response, innovations have been made, such as applying technology in care (e.g., robots and artificial intelligence-assisted systems), community-driven programs to integrate older adults into social activities, and continued investment in healthcare infrastructure.

Considering the broader trends in Japan regarding healthcare and welfare, Toyonaka City provides a unique perspective on urban healthcare management in the face of demographic challenges. As the city continues to adapt and innovate, it serves as a case study for other global urban centers facing similar challenges.

### 2.2. Community Social Workers

CSWs operate within the larger social work framework, but have a distinct focus on the well-being and needs of specific communities [15,16]. These professionals directly engage with local populations to understand their unique challenges and develop interventions to address these issues. CSWs often serve as bridges between vulnerable populations and the services or resources that can assist them. Their responsibilities typically include community outreach, advocacy, crisis response, and policy development. Conducting assessments and collaborating closely with other local organizations ensures that community members can access necessary services such as housing, healthcare, education, and employment opportunities. One of the hallmarks of community social work is its emphasis on empowerment. Rather than merely providing services, CSWs aim to empower individuals and groups within the community to take charge of their well-being and foster resilience and self-sufficiency.

### 2.3. Data Collection

The primary sources of data for this study were Japanese comics based on the real-life experiences of CSWs. Japanese comics are commonly read by young to middle-aged people in Japan. These comics, which constitute visual narratives, offer unique insights into the approaches and strategies employed by CSWs to solve social issues in communities. The main reason for using comics as a data source was to deal with issues of privacy and stigma associated with social isolation and disparities in society. By discussing the data collection with CSWs and the chief researcher (RO), the secondary materials of CSWs, based on the real work of CSWs, were used as data for this study.

A purposeful sampling method was employed to select comics that specifically focused on CSWs’ practices and experiences. The inclusion criteria were as follows: (1) the comic should be nonfictional and based on a real-life account of a CSW, and (2) it should depict specific situations or challenges faced by the CSW and their approach to resolving or navigating them. The potential sample was reviewed by CSWs in Toyonaka, who confirmed whether each comic described true CSW stories. Six comics met the inclusion criteria and were included in this study. The content of each comic’s dialogue, its artwork, and descriptions of each situation in which CSWs worked were transcribed for analysis.

### 2.4. Analysis

Initially, RO conversed with CSWs regarding their concrete work for 60 min; this session was not recorded because it contained various sensitive client issues that could not be used for the study analysis. In this discussion, RO delved deep into CSWs’ working conditions. Second, RO participated in a CSW training workshop in Toyonaka. The workshop lasted two days: the first day focused on the theory underpinning community social work and the second focused on observing CSWs’ real-life practices in Toyonaka. RO’s experiences were shared with YN and CS to establish the same context within the research team.

Reflective thematic analysis was chosen to analyze the comics’ content (i.e., dialogue and descriptions). Braun and Clarke’s six-step approach to thematic analysis was adopted, which offers a systematic framework for identifying, analyzing, and interpreting patterns of meaning within the data [17,18]. In the initial stage of familiarization with the data, two researchers (RO and YN) engaged in multiple readings of the comic transcripts, noting down initial ideas. To generate the initial codes, RO and YN conducted systematic coding across the entire dataset, generating relevant codes for the data pertaining to the research question. Through discussions between RO and YN, codes were sorted into potential themes, and all relevant coded data extracts were collated within the identified themes. Thereafter, the themes were reviewed in relation to the coded extracts and the entire dataset. In case of any conflicts between researchers, the themes were refined, split, combined, or discarded.

To define and name the themes, each theme was refined in terms of its scope and focus. Clear definitions and names were developed for each theme. RO, YN, and CS discussed the content until consensus was reached. The clarified themes were subsequently shared with CSWs in Toyonaka City, and their opinions were collected to revise the themes. RO has presented the findings using vivid and compelling examples, emphasizing the relationship between the analysis, research question, and previous literature. Data were managed using NVivo 11.

#### Reflexivity

The study results are based on interactions between the researchers and data. The members of the research team possess diverse expertise and perspectives on rural medical education. RO, a family physician and medical teacher, graduated with a master’s degree in medical education and family medicine, and has experience working, providing education, and conducting research in rural contexts. YN, a family physician, has experience working in rural communities and hospitals, and conducts qualitative research on social isolation in rural communities. CS, a medical educator and professor at a medical university, graduated from medical university and specializes in community healthcare management and education. To minimize potential bias, the research team exercised caution when discussing the findings of individual data analyses. Alternative viewpoints were explored while inferring the meaning of the data.

### 2.5. Ethical Consideration

The Unnan City Hospital Clinical Ethics Committee approved the study protocol (No. 20230011).

## 3. Results

### 3.1. Thematic Analysis Results

The total number of pages in the comics was 505. Transcripts of the comics as Microsoft Word documents totaled 63 pages. After analyzing and discussing all transcript pages, we identified four themes regarding the specific attitudes, strategies, and approaches of CSWs in Japan (Table 1): the core value of respect for professionalism and dedication to vulnerable people, personalized support oriented toward person-centered suffering, the promotion of inclusive community engagement, and connecting and reorganizing communities to promote inclusive societies. These four themes were stratified and connected to visualize how CSWs effectively apply them in their work to build inclusive societies (Figure 1).

### 3.2. Core Value of Respect for Professionalism and Dedication to Vulnerable People

For CSWs, deepening their understanding of social systems is crucial to addressing inclusivity and other social issues. CSWs update and use their sociology-related knowledge, which serves as the backbone of their activities. Through these experiences, CSWs enrich their understanding of social inclusion, as well as the knowledge and skills needed to serve vulnerable groups. The following comic excerpt regarding CSWs’ reflections on child poverty, which hinders their growth and exacerbates their isolation, demonstrates this idea:

“We must understand the current social systems that create poverty. There are three types of child poverty. The first is economic poverty, which arises from income-related challenges… The second is relational poverty. While economic poverty can be somewhat addressed with available support when sought, as living conditions become tougher, connections weaken… The third type is cultural poverty, which arises when individuals only understand their own teachings and therefore don’t cook… This can contribute to a cycle of poverty. This is why community dining spaces for children hold great significance, as they aim to expand child-rearing practices and familial relationships”.(Comic 5)

Being consistently available for community members and responsive to their consultations is also important. CSWs should be considered gate openers for all vulnerable individuals in communities. To increase trust in CSWs, quick responses to inquiries are essential, as such attitudes encourage citizens’ trust in CSWs and interest in social issues. CSWs consider open-mindedness and prompt responsiveness vital for their effectiveness, as seen in the following excerpt regarding isolated and vulnerable older women with hoarding tendencies. CSWs described the importance of quick responses, leading to the gradual motivation of citizens’ involvement in community issues:

“CSWs are frequently present in local areas, providing them many opportunities to identify daily issues faced by residents. They often initiate support based on matters they find concerning. Although CSWs must independently address some issues, it’s even more beneficial for community members to identify problems and collaborate on solutions. Such a collaborative approach strengthens the community’s capabilities and resilience, and CSWs therefore, immediately respond to community inquiries. Moreover, even when offering support amid bureaucratic challenges, a determined and unwavering attitude is required”.(Comic 1)

CSWs maintain a steady trust in others and believe that community engagement will eventually improve. When initiating dialogues, CSWs are mindful to recognize the layers in the discussion, create relationships by engaging in others’ daily interests, and move gradually to the target issues. The following excerpt from Comic 1 of a conversation between a CSW, neighbors, and a woman with hoarding tendencies shows their interest in vulnerable people’s habitual activities, which leads to effective conversations:

[Responding to the neighbors’ demands for the exclusion of one person with hoarding tendencies]

CSW: You’ve been worried about the situation for a long time, haven’t you? We also want to support Hana-san, so please help us.

Neighbors: Uh, okay.

CSW: Thank you again in advance. I can’t advise on exclusion, okay? Please check on them from time to time. If something happens, I’ll be there right away (Figure 2).

CSW: Oh, is that you, Hana? [She lives in a hoarder house because of lack of family support and has lost the motivation to consult] Hello, Hana. What brings you here?

Client: Ah, it is you. Always lively, aren’t you? Do you like hamburgers too?

CSW: This one’s just like yours, Hana [Showing a bag given as a gift at a hamburger shop].

Client: I got this one in front of the station.

CSW: I go there too. It’s always crowded, isn’t it?

Client: Yeah. When it gets crowded, it feels comforting.

CSW: All right, I’ll visit again then [After this scene, the CSW visits Hana’s hoarder house and resolves the issue] (Comic 1) (Figure 3).

Thus, CSWs try to narrow the mental distance between themselves and clients using ordinary life activities, which gives them the chance to manage clients’ difficulties, in this case, isolation and hoarding.

CSWs prioritize advocating for the disadvantaged and vulnerable, helping to ensure their access to social security. CSWs consistently approach those in distress openly in order to engage with them and create deep connections. CSWs are moved and fulfilled by changes observed in the disadvantaged, which motivates them to persevere with their work. For example, a CSW stated the following regarding a homeless woman suffering from isolation and the lack of a will to live, hindering her ability to improve her situation:

“For those who have long been obstructed by many and who have persisted, building a relationship of trust is no trivial matter. Thorough respect for an individual is required. It’s crucial to value the individual’s feelings and construct support based on their self-determination. However, in many cases, support is built on a provider-centric premise, prioritizing services. A deep respect for individuals will, in turn, quickly create relationships with them”.(Comic 2)

### 3.3. Personalized Support Oriented toward Person-Centered Suffering

CSWs prioritize providing immediate support in response to residents’ concerns and inquiries. CSWs focus on individual problems, value relationships, and consider others’ feelings when advocating for them. For example, in the case of a man who lost his job due to economic deflation and could not consult with the local government or his neighbors, CSWs tried to reignite his motivation and advocated for him. Reflecting on the present system challenges in supporting such a man, a CSW stated the following:

“The challenges faced by those facing financial hardships are among the most difficult for individuals to signal for help. Given the prevailing notion of personal responsibility, seeking consultation at City Hall can be a significant hurdle. In other words, seeking help can make one feel humiliated; so offering support with an understanding of this sentiment is essential. I often encounter individuals who become despondent and are on the verge of giving up on their lives. However, even if the person says that they are fine, as supporters, we must not give up on those close to giving up on life. Financial hardship can happen to anyone, and society needs to adopt a perspective that aims to rebuild lives, starting with welfare support”.(Comic 4)

CSWs transform individual issues into collective concerns by reaching out to various agencies. CSWs have accepting attitudes, understand layered conversations, further dialogue, candidly express their feelings and seek to empathize with others. In cases of hoarder houses, homelessness, and job loss due to brain stroke, CSWs are often initially refused by vulnerable people. However, they gain their trust by expressing their understanding of the difficulty of gaining social support and showing empathy and motivation for their recovery. One CSW, referring to a homeless woman who eventually obtained a house and brought rice to CSWs and other supporters in gratitude, stated the following:

“Even if someone happens to be homeless when we meet them, by reflecting on the various stories that have shaped their life up to that point, we can help draw out their strengths. Making curry rice reminded her of her life journey, and the look on her face when she feels needed by others hints at times when she truly shined”.(Comic 2)

CSWs often employ a collaborative approach to tailor their assistance to each individual’s challenges and consider which support methods best suit each person in need. They are mindful of the beneficiaries during problem resolution, recognizing that each person requiring support has a unique background, and are also conscious of establishing support systems focusing on disadvantaged groups. The comics showed how CSWs help vulnerable people by using both already-established support and new collaborations with various organizations. Reflecting on the difficulty of finding new work opportunities for an unemployed man having trouble with neighbors, one CSW stated the following:

“It’s easy to say, ‘connect them to administrative services’, but for those with complex pasts, there’s a significant fear of their histories being probed into when utilizing these services. Therefore, taking an active role and moving with individuals on their journey through what we call the ‘co-traveling’ support method is effective. Instead of judging whether individuals can use a service, it is essential to support them with the understanding that they are the primary stakeholders in their own lives”.(Comic 3)

### 3.4. Promoting Inclusive Community Engagement

CSWs strive to involve community residents in the support process and actively engage them in dialogue throughout the problem-solving process. They inspire motivation while preserving the community’s unique context, and urge members to participate in activities. For example, CSWs learn about people’s backgrounds and the community’s history, and respect what citizens value and take pride in. This leads to mutual understanding and motivation for people living alongside vulnerable individuals. Reflecting on the community’s tendency toward vulnerability and possibilities, and insisting on the importance of believing in community potential, a CSW stated the following:

“Communities inherently have both exclusive and inclusive characteristics. Particularly in local areas, the stance of community leaders—whether they adopt an exclusionary or inclusive perspective—significantly affects the lives of those facing challenges and requiring social support. By integrating a methodology through which community leaders learn through individual challenges, creating an inclusive environment ensures a stable living foundation and plays a crucial role in subsequent support scenarios. Believing in the power of community and adopting a collaborative, supportive stance can truly transform the community”.(Comic 2)

CSWs observe exclusionary attitudes among residents and counteract these by promoting inclusivity, which they aim to enhance through problem solving; by addressing one issue, they can elevate the community’s overall inclusivity. Recognizing the importance of disseminating increased inclusivity within the community, CSWs share case studies with residents to enhance their understanding of their community. Thus, they frequently use these stories to discuss one issue in depth with citizens and community stakeholders, and describe social issues as personal and community issues. Through this process, citizens and stakeholders are gradually motivated to approach social issues as collective concerns. For example, one CSW, reflecting on the importance of sharing vulnerable people’s issues and regarding residents’ consideration of these issues as community issues, expressed the following:

“Often, individuals perceived as ‘troublesome’ in a community are actually grappling with underlying challenges. Behaviors exhibited by those with hoarding disorders, dementia, or other anxieties often appear problematic to neighbors. Community members can gain insight into their circumstances by focusing on the root issues faced by these individuals. This fosters a perspective of inclusion rather than exclusion. Specifically, it is crucial to contemplate how to continually support these individuals in the communities they know, with CSWs playing a pivotal role in accompanying and assisting them in their journey”.(Comic 6)

Through concrete activities and interactions with CSWs and community members, those receiving support improve, transitioning from being supported to being supporters. CSWs also create new roles for residents, advocate for initiatives, and make individualized suggestions to improve their social participation and ensure diverse societal involvement. For example, a girl in poverty could regain her motivation, go to school, and pay it forward to teach other students in challenging situations in the community through the support of CSWs and motivated citizens who opened community dining spaces for poor children. Reflecting on the poverty issues present among children and possible solutions, a CSW said the following:

“It is precisely because one is cherished that they can evolve into someone who supports others. Indeed, aligned with the theme of community coexistence, those who feel valued by society can learn, regain confidence, and develop profound respect for every individual who wishes to support others. Such transformations require time and space. As families have become smaller and influence from individuals outside immediate family members diminishes, now more than ever, community dining spaces for children serve as crucial junctions connecting the community, schools, and homes”.(Comic 5)

### 3.5. Connecting and Reorganizing Communities to Promote Inclusive Societies

CSWs recognize the importance of connecting individual problems to broader societal issues; for example, they take a single local issue and magnify it so that it concerns the entire community, and thereby determine solutions. From individual to systematic solutions, CSWs support the creation of new relationships, which they use to create new foundations for disadvantaged people. CSWs frequently share personal difficulties with citizens and stakeholders, and create opportunities to discuss social issues. Initially, this discussion is often superficial; however, by sharing various cases of vulnerable people’s issues and how they affect the entire community, they gradually gain motivation to establish more effective social systems. As such, the following was stated by one CSW:

“We must address the very structural issues in society that make it difficult for those without familial support to live comfortably. Recognizing that individuals often share challenges, it’s imperative to understand and communicate these issues to local governments and communities. What’s called for is a proactive approach to resolving these challenges”.(Comic 2)

Within this process, CSWs improve their knowledge of modern societal issues, enabling residents to understand that their local problems are not unique but community-level challenges. Accordingly, CSWs work to increase inclusivity on a community-wide scale by connecting unresolved societal problems to community empowerment, striving to establish a system that embraces different kinds of people. New social changes caused by the COVID-19 pandemic and social disparities caused by critical poverty and an increasing number of single mothers complicate these issues. By continuously learning about such issues, CSWs consider new ways to restructure inclusive and mutual support systems by collaborating with entire communities. In addition, they consider vitalizing potential human resources in communities, such as experienced older adults. A CSW stated the following:

“I believe that it is essential for every individual in a community to provide both support and be supported. Regardless of whether it’s individuals with dementia sharing stories from the past, ingenious solutions during mask shortages amid the COVID-19 pandemic, war veterans in wheelchairs sharing their wartime experiences, or older adults contributing their professional experiences to local volunteer activities, older adults have more to offer than merely being care recipients. Their vast wisdom and experience should be harnessed for societal benefit. While receiving caregiving support, their personal experiences can greatly contribute to the community”.(Comic 6)

By fostering new relationships with disadvantaged people throughout the community and solving social issues, CSWs promote community growth, including its capacity for inclusivity. To determine solutions for unresolved societal challenges, CSWs connect various related agencies, organically link and reorganize diverse social resources, and advocate for better community capacity. The comic stories show that CSWs detect as many potential resources in the community as possible and connect them effectively to facilitate the solving of various social issues. In this process, CSWs motivate community members and vitalize the community as a whole by collaborating with potential local resources. CSW hope to establish mutually supportive and inclusive societies in which everyone can be supported at any time, as seen in the following:

“The perspective on promptly identifying and approaching individuals who cannot send out SOS signals is critical. Many within the population experiencing genuine distress cannot voice their struggles. Additionally, with the transition from welfare service placement to contracts, there’s a need for community-building to identify individuals who refuse contracts or are unable to send SOS signals. Beyond building supportive structures such as local and oral welfare committees, future efforts must also focus on reaching out to individuals in apartments or those who do not participate in neighborhood associations. Strategies such as monitoring systems targeting those outside neighborhood associations and ‘rolling operations’ are essential. Collaborative networks involving electricity, gas, water utilities, pioneering businesses, newspapers, delivery services, and various other sectors play a vital role in ensuring that no individual is left unattended. Diverse methods are required to ensure that no one falls through the cracks”.(Comic 6)

## 4. Discussion

This study sheds light on the unique strategies adopted by CSWs in Japan to combat social isolation among vulnerable people in an aging society. Through thematic analysis, we distilled their methods into four main themes regarding their offering of a holistic approach to community inclusivity.

Regarding the core value of respect for professionalism and dedication to vulnerable people, CSWs in Japan manifest an ethos of unwavering commitment. A previous study showed that commitment to marginalized communities is pivotal to ensuring equity in service provision [19]. Within this spectrum, CSWs’ emphasis on the necessity of understanding social systems resonates with the findings of another study that posited that a deep grasp of societal structures is paramount to affecting systemic change [20]. This foundational understanding facilitates the tailoring of interventions and ensures that CSWs are poised to serve as authentic voices for the marginalized [21]. Their consistent availability and proactive approach underscore their innate dedication [22]. Such dedication often transcends mere professional obligations, indicating a deeply rooted intrinsic motivation to foster change [23]. Furthermore, the gradual relationship-building strategy adopted by CSWs finds parallels in previous research, which highlights the importance of trust in the social work paradigm and elaborates that the slow nurturing of relationships is imperative for establishing this trust, further emphasizing that it forms the bedrock of meaningful and impactful social work [24,25].

Concerning personalized support oriented toward person-centered suffering, this theme combines the fundamental principles of social work with the concept of client- or person-centered therapy [26]. Capturing the essence of modern social work, person-centered care hinges on the belief that individuals can inherently understand and alter their situations [27]. CSWs exemplify a nuanced, patient-oriented, and intuitive methodology in this context. As previous research on social care in East Asia has shown, understanding the unique complexities of each individual’s challenges is paramount for crafting tailored interventions [28]. This philosophy is brought to life when CSWs shift their personal issues to broader collective concerns. This approach aligns with a previous study’s findings, determining that fostering community support often entails reframing individual challenges as shared experiences [12,29]. This transformative approach transcends merely offering solace; it instigates community cohesiveness and mutual support [30]. By veering away from isolating problems and advocating for collective solutions, CSWs ensure individualized care and propagate a reinforced sense of community responsibility and involvement, a sentiment echoed by previous studies on the role of community in holistic well-being [31].

Regarding the theme of exclusive to inclusive community engagement, CSWs emerge as linchpins that catalyze this transformative shift. As posited by previous studies, the cornerstone of progressive community development lies in dismantling exclusive paradigms in favor of more inclusive and cohesive frameworks [32,33]. In this context, CSWs are not mere troubleshooters; their visionary interventions align with previous research that has emphasized the importance of proactive strategies in encouraging genuine community growth [34]. One of the most salient facets of CSWs’ approach is their emphasis on metamorphosing the roles of community members from passive recipients of support to active contributors. Previous studies have highlighted a similar paradigm shift in empowerment models, noting that transitioning from being aided to aiding fosters a profound sense of purpose and belonging [35]. This transformation, championed by CSWs, does more than empower individuals. It creates a sustainable community ecosystem, described as a “symbiotic cycle of support”, wherein individuals, bound by mutual respect and empathy, continuously uplift and champion one another, ensuring the community’s resilience and holistic growth [7].

CSWs’ intricate and holistic approach is prominent in connecting and reorganizing communities to promote inclusive societies. Consistent with previous research, the ability to extrapolate individual issues and portray them as collective challenges is a transformative strategy that fosters collective ownership and solidarity [36,37]. Reframing catalyzes the galvanization of communities, motivating them to rally around shared challenges and drive systemic change. Central to this effort is CSWs’ role in fostering community participation. According to a previous study, genuine community engagement stems from a unified approach in which the lines between personal and collective concerns blur, ensuring widespread involvement in problem solving [38]. CSWs adeptly cultivate this environment.

Furthermore, their nuanced approach to leveraging diverse social resources, as noted by other social work studies, amplifies the community’s capacity to address issues and maximizes community assets for sustained impact [39]. Crucially, CSWs not only act as facilitators, but also as educators. By channeling these resources and perspectives, they instill a profound understanding among community members regarding their shared responsibility in proactively addressing and resolving pervasive social issues, an ethos underscored in their discourse on shared community responsibilities [40].

The current study’s findings underscore the profound impact CSWs have on fostering inclusive societies. Their multifaceted approach, ranging from individual support to community engagement, reinforces the importance of holistic interventions in social work. Their methods combat social isolation and promote community growth and resilience. Additionally, this study provides invaluable insights for global social work practices. Although this study focuses on Japan, the strategies and approaches of CSWs discussed here can serve as a blueprint for social workers worldwide, as they emphasize the universality of the principles of trust, inclusivity, and community engagement. Future research should explore the applicability of these strategies in various cultural and societal contexts to provide a comprehensive toolkit for global social workers.

Despite offering valuable insights, this study had several limitations; chief among them is its reliance on comic transcripts, which, despite their narrative richness, might not capture the full intricacies of real-life CSW experiences in Japan. The study’s scope, limited to 63 pages of transcripts, may not represent broader CSW practices nationwide. The inherent subjectivity present in thematic analysis, the cultural context of Japan, the absence of data triangulation, temporal constraints, and potential selection biases in comics may also have influenced the study’s findings. Recognizing these constraints is essential for accurately contextualizing the research outcomes and framing future investigative avenues.

## 5. Conclusions

This research, undertaken to comprehend CSWs’ strategies and approaches to alleviating social isolation in Japan, has illuminated several themes: the importance of professionalism and unwavering dedication to vulnerable populations; the emphasis on personalized, individual-centric support; the transformational role of CSWs in moving communities from exclusiveness to inclusiveness; and their capacity to weave isolated issues into a larger community fabric, fostering collective responsibility and action. The narratives drawn from the comic transcripts underscore CSWs’ multifaceted roles as advocates, facilitators, educators, and connectors, cementing their place as pivotal agents in creating inclusive societies. However, it is imperative to interpret these findings while considering the research limitations. As societal landscapes evolve, the strategies and ethos of CSWs will undoubtedly adapt, emphasizing the need for ongoing research in this domain to keep pace with the changing dynamics. This study serves as a foundational step, spotlighting the commendable work of CSWs in Japan, and lays the groundwork for more in-depth and expansive inquiries in the future.

## Figures and Tables

**Figure 1 geriatrics-08-00113-f001:**
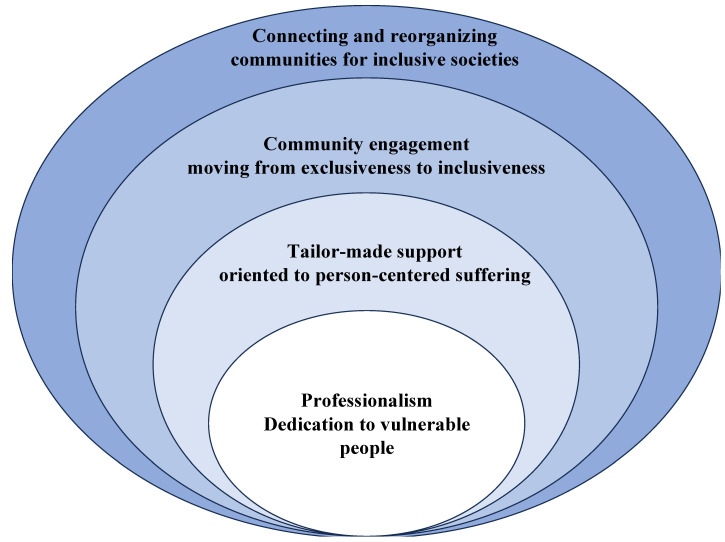
Concept map of community social workers’ efforts to promote inclusive societies.

**Figure 2 geriatrics-08-00113-f002:**
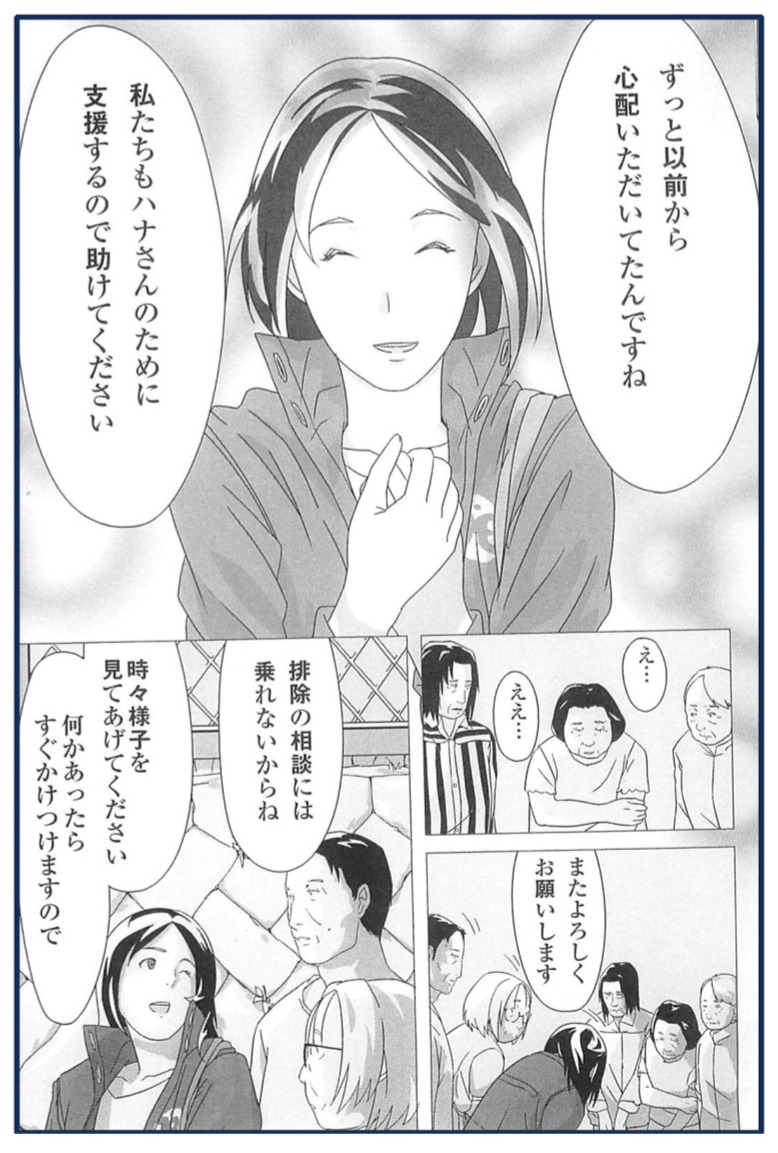
Interaction between CSW and neighbors of a woman with hoarding tendencies.

**Figure 3 geriatrics-08-00113-f003:**
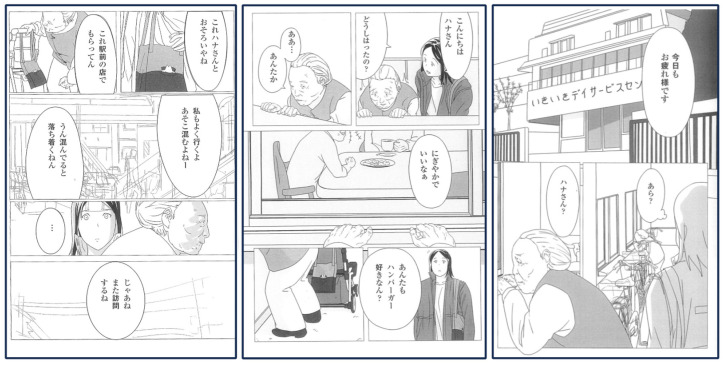
Interaction between CSW and Hana living in a hoarder house.

**Table 1 geriatrics-08-00113-t001:** Results of the grounded theory approach.

Theme	Explanation
Core value of respect for professionalism and dedication to vulnerable people	Improving their understanding of social systems is key for CSWs. They continually enhance their comprehension of social systems to address inclusivity and social issues. When approaching a community, it is vital that they are available and responsive to any consultation without evasion. They maintain an unwavering trust of others, believing those they engage with will inevitably change over time. When initiating a dialogue, they are mindful of recognizing layers within the conversation, building relationships by leveraging others’ interests and gradually transitioning to the issues they seek to resolve. CSWs place great emphasis on becoming the voice for the disadvantaged. When these individuals cannot express themselves, CSWs advocate for them, ensuring they can easily access social security. They always attempt to open up the hearts of those in distress and consistently approach them openly, which allows them to connect deeply and continue their support. CSWs feel both moved and fulfilled by positive changes for disadvantaged groups, which motivates them to persevere in their work.
Personalized support oriented toward person-centered suffering	CSWs prioritize being immediately supportive in response to residents’ concerns and are swift in addressing their consultations. They are interested in individual problems and cherish their connection with others, considering others’ feelings and advocating for them. Through dialogue with various related agencies, CSWs transform individual issues into collective concerns. They demonstrate an accepting attitude, understand layers of conversation, advance dialogue, candidly express their feelings, and patiently approach any reaction from the other side. By providing tailored assistance to each individual’s challenges, CSWs consistently embody a collaborative support approach. They continuously contemplate the support methods best suited for each person in need. They are mindful of recipients during problem resolution, recognizing that each person requiring support has a unique background; CSWs consistently practice this consideration throughout their assistance efforts. They are conscious of establishing a support system centered on the disadvantaged.
Promoting inclusive community engagement	CSWs are conscious of involving community residents in the support process, actively engaging in dialogue with them throughout the problem-solving process. They work to inspire motivation without damaging the community’s circumstances and urge them to participate in actual activities. When observing instances of exclusionary attitudes among residents, they make efforts to promote inclusivity. They aim for enhanced inclusivity through problem solving, realizing that by addressing one issue, they can improve overall community inclusivity. Recognizing the importance of sharing inclusivity growth within the community, CSWs deepen residents’ understanding of their community by sharing case studies. Through actual activities, they help those receiving support to transition from being supported to being supporters themselves via interactions with CSWs and community members. CSWs also push forward initiatives to create new roles for residents, continuously suggesting methods for individualized social participation tailored to their life stages, ensuring diverse societal involvement.
Connecting and reorganizing communities to promote inclusive societies	CSWs recognize the importance of connecting individual problems to broader societal issues, taking a single local issue and magnifying it to a community-level concern, thereby structuring solutions. From individual resolutions to systematizing solutions, CSWs support establishing new relationships to create a new foundation for the lives of the disadvantaged. Within this process, CSWs expand their knowledge of modern societal issues, reconstituting social resources to help residents understand that the problems they perceive are not unique but rather challenges the community should address. By linking unresolved societal problems to community empowerment, CSWs work to increase inclusivity on a community-wide scale, striving to establish a system that can embrace diverse individuals. By supporting the establishment of new relationships for the disadvantaged throughout the community, CSWs promote community growth, including its capacity for inclusivity, via solving social issues. In their quest to find solutions to unresolved societal challenges, CSWs connect various related agencies, organically linking and reconstituting diverse social resources and continuing to advocate for enhanced community capacity through seamless support for the disadvantaged.

## Data Availability

The datasets used and/or analyzed in the current study may be obtained from the corresponding author upon reasonable request.

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
