# Peer review of "Mutually Supportive and Inclusive Societies Driven by Community Social Workers in Japan: A Thematic Analysis of Japanese Comics"

_geriatrics, 2023, doi:10.3390/geriatrics8060113_

Round 1
Reviewer 1 Report
Comments and Suggestions for Authors
Author Response
Dear Authors,
The manuscript titled “Mutually supportive and inclusive societies driven by community social workers in Japan: A thematic analysis of comics”
The abstract should be revised according to the guidelines of the journal.
Response:
Thank you for your feedback. We have revised the abstract according to the journal guidelines as follows.
Abstract: Social isolation is a growing concern worldwide, particularly within aging populations. This study elucidates the specific attitudes, strategies, and approaches of community social workers (CSWs) in Japan as they work toward alleviating social isolation and building inclusive communities. This qualitative study, conducted in Toyonaka City, Osaka Prefecture, Japan, used six Japanese comics as a unique data source narrating real-life stories of social isolation and CSWs’ approaches. Thematic analysis was conducted to analyze content, including systematic coding, theme generation, and refining, while ensuring rigor and reflexivity. The total number of pages of the comics was 505. Transcripts of the comics as Microsoft Word documents totaled 63 pages. Four themes characterizing CSWs’ strategies were revealed: 1) core values of professionalism and dedication; 2) personalized support oriented toward person-centered suffering; 3) community engagement, transitioning from exclusiveness to inclusiveness; and 4) connecting and reorganizing communities for inclusive societies. In Japan, CSWs employ multifaceted strategies to combat social isolation and foster inclusive communities. Their dedication, personalized support, community engagement, and capacity to reorganize their communities contribute to their pivotal role. This study provides a foundation for understanding CSWs’ work and paves the way for further investigation of their evolving role in creating inclusive societies.
(Lines 12 to 27)
Introduction
The introduction and objectives are unclear due to the abstract.
Response:
Thank you for your feedback. We have made revisions and additions to the objectives of the study in the introduction and abstract.
“This study elucidates the specific attitudes, strategies, and approaches of community social workers (CSWs) in Japan as they work toward alleviating social isolation and building inclusive communities." (Lines 13 to 15)
“Gaining clarity on the concrete approaches of CSWs can provide critical insight into strategies to alleviate social isolation [13]. This study examines the specific attitudes, strategies, and approaches of CSWs in Japan in their battle against social isolation. Through this exploration, we hope to offer a comprehensive understanding that can guide future interventions and policies.” (Lines 57 to 61)
Keywords: should be ordered according to the journal's instructions, e.g. research area is placed last. Please limit your use to 4 to 5 words because they are too many.
Response:
Thank you for your feedback. We have reduced the number of keywords.
Why do authors only describe (n=6) once?
Response:
We have revised the relevant part in the abstract as follows.
“This qualitative study, conducted in Toyonaka City, Osaka Prefecture, Japan, used six Japanese comics as a unique data source narrating real-life stories of social isolation and CSWs’ approaches." (Lines 15 to 16)
Japanese manga (line 15, 119) Could authors please elaborate?
Response:
We have changed the word “manga” to “comic” in the manuscript.
The authors should go into more detail and discuss the findings and results of the research in the Discussion Section. This would make a significant enhancement to the paper.
Response:
We comprehensively checked and revised the discussion section, which includes solutions for vulnerable people in developed countries.
The bibliography is short. Most publications are recent and very recent and provide adequate support for documentation and comparative presentation of results.
Response:
Thank you for your feedback. We have checked the references and updated them.
Reviewer 2 Report
Comments and Suggestions for Authors
This study employs a unique approach to assess depiction of social isolation in commonly used media, with real life stories being used.
It may be enhanced by providing readers additional information and addressing these issues-
1. The abstract needs to more clearly describe the methodology, including the contribution of CVSWs to selecting the 63 pages.
2. Please provide the population of the region/city and the proportion who are older
3. It is unclear whether the manga were actually being read by older people- please clarify. If the main audience is younger people, please note this as it clearly indirectly impacts on how they perceive aging and support services.
4. 63 pages seems to be only a small sample of the comics, which I imagine are numerous. Over what time frame were these manga written and how many of these, in total, were assessed? You may be aware of analyses of how movies depict older folk and often all movies- thousands in a recent study- are reviewed.
5. You have only given one verbal transcript- the paper could be enriched with more examples, perhaps even as images (translated), as many readers would be unfamiliar with manga.
Author Response
This study employs a unique approach to assess depiction of social isolation in commonly used media, with real life stories being used. It may be enhanced by providing readers additional information and addressing these issues-
- The abstract needs to more clearly describe the methodology, including the contribution of CVSWs to selecting the 63 pages.
Response
We have added the description of total pages of Microsoft word document as follows.
“Transcripts of the comics as Microsoft Word documents totaled 63 pages.”(Lines 170-171)
- Please provide the population of the region/city and the proportion who are older
Response
We have added the description of the population of the region/city and the percentage of older adults as follows.
“In 2020, 25.7% of the population comprised over 65-year-old people.” (Lines 77 to 78)
- It is unclear whether the manga were actually being read by older people- please clarify. If the main audience is younger people, please note this as it clearly indirectly impacts on how they perceive aging and support services.
Response
We have added the description of target population of Japanese comics as follows.
“Japanese comics are commonly read by young to middle aged people in Japan.” (Lines 110 to 111)
- 63 pages seems to be only a small sample of the comics, which I imagine are numerous. Over what time frame were these manga written and how many of these, in total, were assessed? You may be aware of analyses of how movies depict older folk and often all movies- thousands in a recent study- are reviewed.
Response
The total number of pages in the Japanese comics is 505. Therefore, in our study, we have included the total page count of the transcript, which was generated as a Microsoft Word document.
“The total number of pages of the comics was 505. Transcripts of the comics as Microsoft Word documents totaled 63 pages.” (Line 171 to 172)
- You have only given one verbal transcript- the paper could be enriched with more examples, perhaps even as images (translated), as many readers would be unfamiliar with manga.
Response
We have included two images depicting Japanese comics in our research paper. However, we were unable to translate the text within the images into English due to permission-related restrictions. Therefore, we have provided the original version of the images and included translated excerpts in the main text.
Round 2
Reviewer 1 Report
Comments and Suggestions for Authors
I suggest accepting. It was a good evolution in writing the study. Congratulations.